# Precision multiplexed base editing in human cells using Cas12a-derived base editors

Anabel Y. Schweitzer [1,2], Etowah W. Adams[1,2], Michael T. A. Nguyen [1,2], Monkol Lek [3] & Farren J. Isaacs [1,2,4] ✉

Base editors enable the direct conversion of target nucleotides without introducing DNA double strand breaks, making them a powerful tool for creating point mutations in a human genome. However, current Cas9-derived base editing technologies have limited ability to simultaneously edit multiple loci with base-pair level precision, hindering the generation of polygenic phenotypes. Here, we test the ability of six Cas12a-derived base editing systems to process multiple gRNAs from a single transcript. We identify base editor variants capable of multiplexed base editing and improve the design of the respective gRNA array expression cassette, enabling multiplexed editing of 15 target sites in multiple human cell lines, increasing state-of-the-art in multiplexing by three-fold in the field of mammalian genome engineering. To reduce bystander mutations, we also develop a Cas12a gRNA engineering approach that directs editing outcomes towards a single base-pair conversion. We combine these advances to demonstrate that both strategies can be combined to drive multiplex base editing with greater precision and reduced bystander mutation rates. Overcoming these key obstacles of mammalian genome engineering technologies will be critical for their use in studying single nucleotide variant-associated diseases and engineering synthetic mammalian genomes.

The development of target-specific and efficient genome engineering technologies has transformed the study and generation of mammalian cell systems, with CRISPR-associated (Cas) nucleases being the most widely used technology. However, Cas-mediated editing outcomes are imprecise as they rely on the introduction of DNA double-strand breaks (DSB), initiating cellular repair processes that ultimately lead to unintended genome modifications[1]. Recently, extensive engineering efforts have expanded the capabilities of Cas nuclease-based genome engineering tools, enabling the precise installation of small insertions or deletions (prime editing[2] and click editing[3]), the introduction of point mutations (base editing[4,5]), and epigenetic modifications into the human genome. Amongst these genome engineering technologies, base editing is unique as it enables direct chemical modification of the

target nucleotide bases and avoids the introduction of DSBs[4]. DNA base-pair modifications are achieved by fusing a deaminase moiety to a catalytically impaired Cas nuclease to form a base editor (BE) protein. Using guide RNAs (gRNA), BEs are localized to specific sites in a genome where they install targeted base-pair conversions at high frequencies. The two main classes of BEs, Cytosine BEs (CBEs)[4] and Adenine BEs (ABEs)[5], mediate C-to-T and A-to-G conversions, respectively. Cas9-derived BEs have successfully been utilized to study and revert trait-associated single-nucleotide variants (SNVs) in mammalian genomes[6].

The ability to reconstruct polygenic mutations underlying complex phenotypes (e.g., cancer) or to engineer synthetic mammalian genomes[7] is hampered by two key limitations of Cas9-based BE

[1]Department of Molecular, Cellular, and Developmental Biology, Yale University, New Haven, CT, USA. [2]Systems Biology Institute, Yale University, West Haven, CT, USA. [3]Department of Genetics, Yale School of Medicine, New Haven, CT, USA. [4]Department of Biomedical Engineering, Yale University, New Haven, CT, USA. ✉e-mail: farren.isaacs@yale.edu

systems. First, the delivery of multiple gRNAs into a single cell required for multiplexed base editing (MBE) has inherent technical limitations. Strategies to overcome this issue include expression of multiple gRNAs from independent promoters or engineered arrays, enabling post-transcriptional processing into individual gRNAs[8,9]. However, in these systems, the necessity of additional, repetitive sequence components leads to genetic instability and larger vector sizes, making them harder to construct and posing delivery burdens[9]. While the delivery of *in vitro*-generated gRNAs or ribonucleoprotein complexes can circumvent the described issues, both approaches are costly, not easily accessible to most laboratories, as not all BE systems are commercially available in purified form, and the efficiency of delivery is limited by the size of the protein moiety of the BE system[10]. The second key limitation is rooted in the processivity of the deaminase, which frequently leads to the conversion of neighboring bases, known as bystander mutations (BMs), in addition to the target base, presenting a challenge in applications where precise nucleotide (nt) modifications are necessary[11]. Although alternative genome engineering technologies like prime editing or click editing exhibit higher editing purity than BEs, they suffer from low overall editing efficiencies[2,3] and require similarly complex gRNA expression strategies as Cas9-derived BEs. Thus, creating many precise edits across a genome remains a defining challenge in mammalian genome engineering, impeding comprehensive studies of complex genotype-phenotype relationships and genome-scale engineering efforts.

Although most published BEs utilize a Cas9 nickase (nCas9), BEs utilizing a catalytically dead Cas12a (dCas12a) protein have been developed and shown to mediate similar or higher editing frequencies when compared to nCas9-derived BE systems[12–15]. Because Cas12a can process gRNA arrays without any accessory factors, a property that is distinct from Cas9, it has been leveraged for multiplexed genome engineering[16,17]. Notably, only one report documents Cas12a-mediated base-editing in human cells at as many as five target sites simultaneously[13].

This study addresses these limitations in base-editing by introducing several advances in mammalian genome engineering. We engineer a LbCas12a-derived BE system to edit up to 15 endogenous target sites in multiple human cell lines, increasing the state-of-the-art threefold. We further leverage the mismatch sensitivity of Cas12a to mitigate BMs at a subset of targeted sites, by truncating the gRNAs and demonstrate that this approach can be utilized in conjunction with array-based gRNA expression (Fig. 1A). Together, these strategies provide key advances in multiplexing and precision of BE to address limitations of current genome engineering technologies and establish a path for more comprehensive studies of complex genotypes in human cells.

## Results

### Evaluation of dCas12a-derived base editors for multiplex base editing

Previous studies reported editing frequencies ranging from 8.3% to 41% when employing dCas12a-derived BE systems to target five genomic loci in human cells simultaneously[8,13]. Since these data were collected using different methods, we first sought to conduct a comprehensive screen of published dCas12a-derived BE systems for MBE. We selected three systems utilizing catalytically dead *Lachnospiraceae sp.* Cas12a (dLbCas12a)[12,14] as the nuclease moiety, and three systems utilizing catalytically dead *Acidaminococcus sp.* Cas12a (dAsCas12a)[14,15]. Of the selected systems, two dLbCas12a-derived systems and two dAsCas12a-derived systems were engineered as CBEs (base editing induced by human APOBEC3A and Cas12a without DNA break 1 (BEACON1), base editing induced by human APOBEC3A and Cas12a without DNA break 2 (BEACON2), enAsBE1.1 and enAsBE1.2, respectively) while one dLbCas12a- and one dAsCas12a-derived system were engineered as ABEs (LbABE8e and enAsABE8e, respectively)

(Supplementary Data 1). To test their ability to drive MBE, we constructed two sets of gRNA expression plasmids, one for CBEs and one for ABEs, in which the human U6 (hU6) promoter drives expression of gRNAs targeting genes that had been successfully edited in previous studies[12,14]. Specifically, we chose to target *RUNX1*, *DNMT1,* and *EMX1* (gRNA IDs R1, DN1, and E1) with the selected CBEs and *CDKN2A*, *VEGFA* and *DYRK1A* (gRNA IDs C2, V1, and DY1) with the ABE systems, either expressing them individually (single gRNA, sg) or as part of a gRNA array composed of two (double gRNA, dg) or all three distinct gRNAs (triple gRNA, tg). Using a published protocol[12], we observed low editing for all tested systems with gRNA R1 or C2 in HEK293 cells (Supplementary Fig. 1), motivating a more detailed investigation into method optimization.

To determine an optimized method for MBE, we tested six different protocols using gRNA R1 and a triple gRNA array composed of R1, R2, and R14 (Supplementary Fig. 2B), that differ in selection regimes (e.g., $2 \mu g/mL$ or $0.5 \mu g/mL$ of puromycin) and outgrowth phase length (e.g., 5, 7 or 9 days post-transfection). Protocols 4 and 6 were designed to feature the most stringent puromycin selection regimes and did not yield any viable cells, likely because cells did not maintain the transfected gRNA expression plasmid for long periods of time. All other protocols led to significantly higher editing frequencies for gRNA R1 expressed as a single or from a triple gRNA array composed of gRNA R1, R2, and R14. Selecting for cells that harbor the gRNA expression plasmid using $2 \mu g/mL$ of puromycin and prolonging the outgrowth phase in protocol 2 to seven days (Supplementary Fig. 2, Source Data File) not only led to a significant increase in editing frequency (29.5% to 69.9% for gRNA R1 expressed from a triple gRNA array), but yielded the most viable cells on day seven. Using this protocol, we first tested the dLbCas12a-derived systems and achieved robust editing of all three CBE and ABE target sites with editing frequencies of up to $39 \pm 5\%$ across all gRNA expression conditions (Fig. 1B, C). Next, we tested the same gRNAs with the dAsCas12a-derived BE systems and only observed modest editing frequencies at one target site (*RUNX1* gRNA R1 for enAsBE1.1 and enAsBE1.2, *VEGFA* gRNA V1 for enAsABE8e), independent of the gRNA expression condition (Supplementary Figs. 3A, B and 4A, B). Interestingly, we observed pronounced differences in editing frequencies of LbABE8e with gRNAs V1 and DY1, depending on their position in the array, and what gRNAs they were combined with. We hypothesized that the %GC content of the 5' or 3' located gRNA might influence the secondary structure of the gRNA array transcript, thus affecting the processing efficiency of dCas12a, and downstream editing frequencies. To test this hypothesis, we constructed double gRNA arrays consisting of either gRNA C1, V1, or DY1, and a non-targeting gRNA with either 30% or 80% GC content. The non-targeting gRNAs were cloned both 5' and 3' of the targeting gRNAs, yielding a total of twelve distinct double gRNA arrays. We found that both the %GC content and the position of the non-targeting gRNA influenced the LbABE8e-mediated editing frequency of the targeting gRNA it was paired with (Supplementary Figs. 3C and 4C). While pairing gRNA C1 with either non-targeting gRNA at any position hindered editing at *CDKN2A* (array dg5-dg8), we observed editing at *VEGFA* and *DYRK1A* with at least two of the tested gRNA combinations. Interestingly, the position of the 30% GC non-targeting gRNA in a double gRNA array with gRNA V1 impacted the observed editing frequency at the *VEGFA* locus (array dg9 and dg11), while combining this gRNA with the 80% GC non-targeting gRNA did not mediate editing (array dg10 and dg12). In contrast, we still observed editing at *DYRK1A* after combining gRNA DY1 with either non-targeting gRNA (array dg13-dg16).

Together, these data indicate that all tested BE systems are capable of processing short gRNA arrays, though editing frequencies vary depending on the employed BE system and the exact gRNA array composition. In our hands, dLbCas12a-derived systems outperformed dAsCas12a-derived systems at the six tested target sites, with CBEs

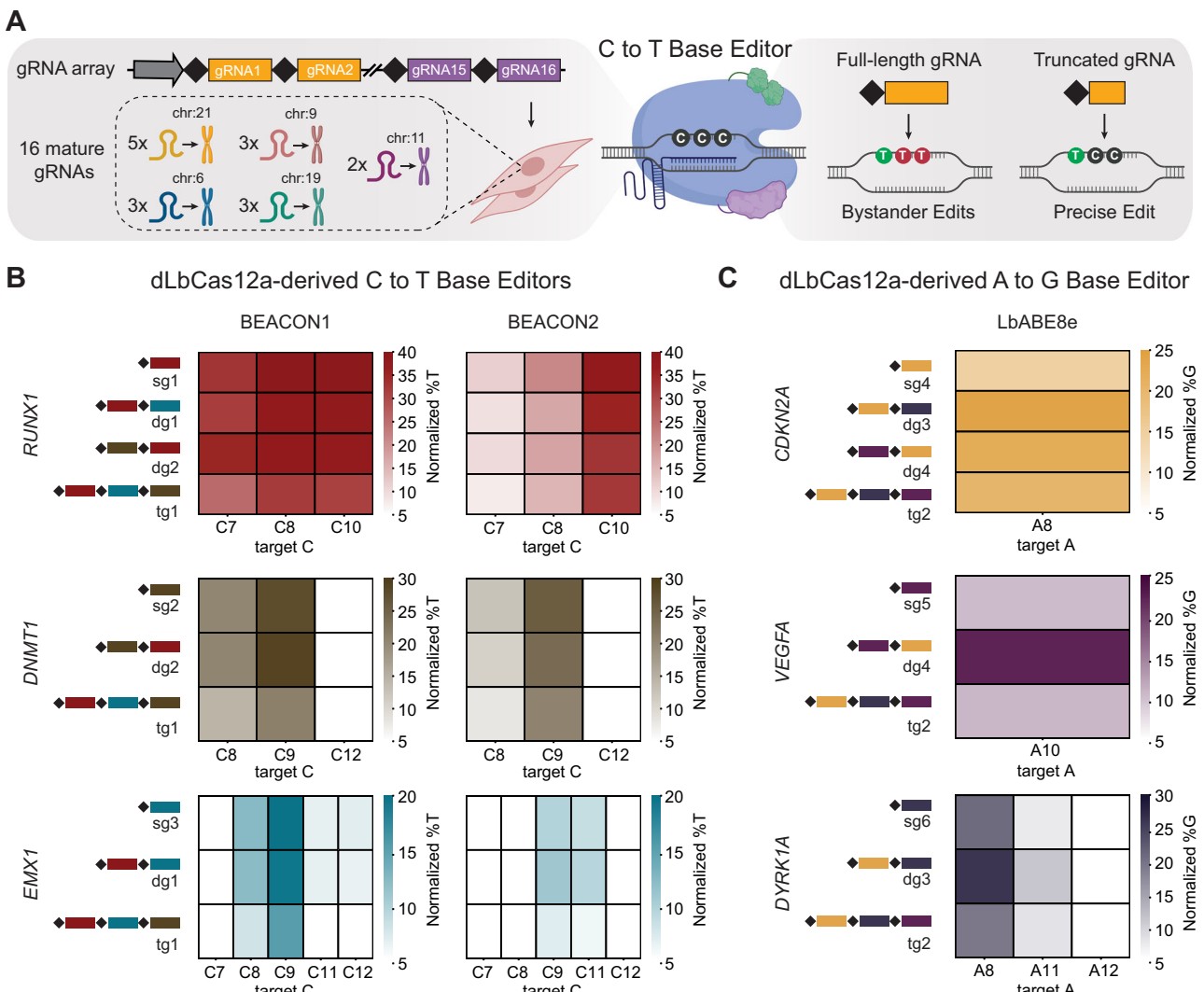

**Fig. 1 | Engineering Cas12a for multiplexed base editing. A** The ability of Cas12a to process gRNA arrays without accessory factors allows for the efficient delivery of many gRNAs into a single cell. Truncated gRNAs can be used to prevent deamination of neighboring bases. Created in BioRender. Schweitzer, A. (2025) https://BioRender.com/z2a6bsw. **B** Comparison of two published dLbCas12a-derived CBE and (**C**), one published dLbCas12a-derived ABE system for MBE in HEK293 cells.

Heatmaps show normalized mean %T/%G values from three independent replicates ($n = 3$). Normalization was performed by subtracting the mean %T/%G values of the nt-ctrl condition from the mean %T/%G values of the experimental condition. Only positions 7–12 for CBEs and positions 8–12 of ABEs are shown, as those correspond to the editing window of the used systems. sg: single guide, dg: double guide, tg: triple guide. Source data are provided as a Source Data file.

achieving slightly higher editing frequencies than ABEs. Based on the high editing frequency of BEACON2, its comparatively narrower editing window, and the previously demonstrated low rate of RNA off-target editing mediated by BEACON2[12], we employed a Sleeping Beauty transposition approach to generate a stable cell line (hereafter HEK293-B2) constitutively expressing this CBE. The HEK293-B2 cell line was used in all subsequent experiments unless otherwise noted.

### Improved gRNA array architecture enables multiplex base editing using dCas12a

Although previous reports demonstrated Cas12a-mediated MBE of five target sites in prokaryotes[18] and human cells[13], we were motivated to explore the possibility of increasing the scale of multiplexing in a single cell. We first aimed to combine gRNAs into a hU6-driven gRNA array following the standard architecture, in which each gRNA sequence is directly preceded by the LbCas12a direct repeat (DR) sequence (Fig. 2A, B). Toward this goal, we validated 14 gRNAs targeting *RUNX1* (Supplementary Data 1) and observed editing frequencies ranging from $25.7 \pm 1.1\%$ to $70.2 \pm 2.4\%$ (Fig. 2C). Expressing these gRNAs using

the standard array architecture mediated editing at 12/14 targeted sites, with efficiencies between $9.9 \pm 2.0\%$ and $51.9 \pm 3.1\%$ for the main target base. We hypothesized that gRNAs that mediate high editing as single gRNAs but not when they are expressed from an array may not be efficiently processed by dCas12a due to undesired RNA secondary structure formation. Thus, we constructed additional versions of the gRNA array, in which neighboring gRNAs were intervened by a four nt synthetic separator (SynSep) sequence. The first version contained a fixed AAAT SynSep, which has been shown to improve array processing in the context of CRISPR activation experiments[19]. The variable SynSep (VarSep) arrays contained four variable nts at each separator position, designed to promote ideal gRNA array folding. With the AAAT SynSep array, we observed significantly increased editing frequencies for some gRNAs (e.g., gRNA2) but also significantly decreased efficiencies (e.g., gRNA14) (Fig. 2C, D and Supplementary Fig. 5). Such negative effects of SynSeps on gRNA performance have not previously been reported and suggest that the incorporation of an AAAT SynSep is not universally applicable to enhance gRNA array processing. Similarly, incorporation of VarSep sequences into the gRNA array had

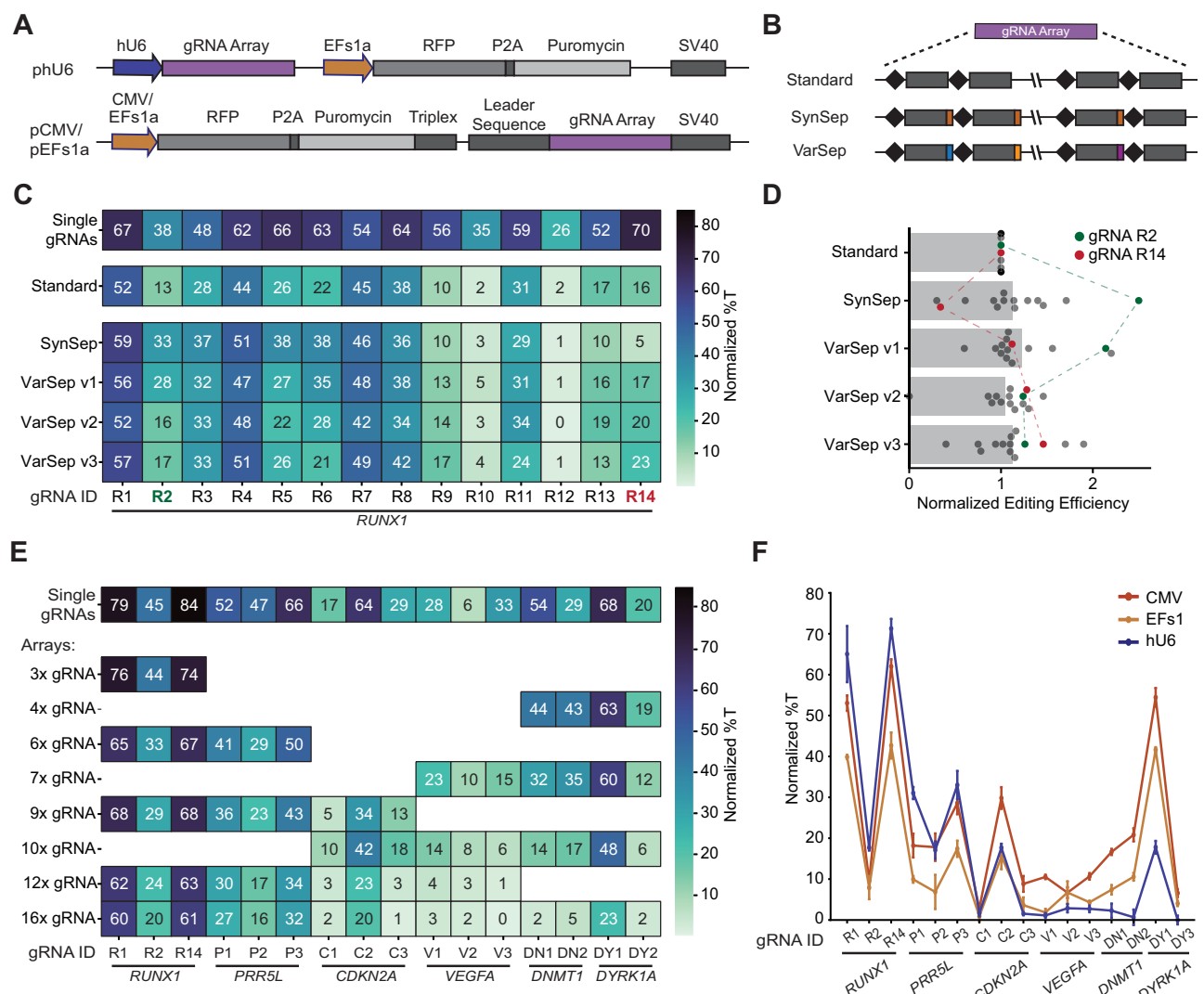

**Fig. 2 | BEACON2 mediates highly multiplexed base editing. A** Schematic of the architecture of the three gRNA expression plasmids used in this study. SV40: Simian Virus 40 termination signal. **B** Schematic of gRNA array architectures used in this study. The standard array consists of the DR and guide sequences and does not include any additional sequence elements. The SynSep array includes an AAAT sequence upstream of each DR sequence, and the VarSep arrays include variable 4 nt sequences upstream of each DR sequence (see Supplementary Data 1). **C** Comparison of different array architectures for BEACON2-mediated multiplex editing of 14 *RUNX1* target sites. **D** Editing frequencies of gRNAs across arrays shown in (**C**), normalized to their efficiency when expressed using the standard array architecture. **E** BEACON2-mediated multiplex editing of 3–16 target sites, across six genes located on five chromosomes. **F**, Comparison of a Pol-III promoter (hU6) and two Pol-II promoters (CMV and EFs1a) for the expression of the standard gRNA array shown in (**D**). All data are the mean ± SD of three independent replicates (*n* = 3). **C**–**F** show normalized mean %T values at the highest edited C for each gRNA as determined based on the single gRNA condition. Normalization was performed by subtracting the mean %T of the non-targeting (nt-ctrl) condition from the mean %T of the experimental condition. Source data are provided as a Source Data file.

diverse effects on the editing frequencies of the associated gRNAs, and in some cases resulted in higher editing frequencies than with the standard or AAAT SynSep array architecture (e.g., gRNA9 and gRNA14), confirming that this effect is separator sequence-independent (Fig. 2C, D and Supplementary Fig. 5).

Motivated by the ability to edit 14 target sites across *RUNX1*, we sought to explore if a single gRNA array could mediate editing of six genes across five chromosomes simultaneously. Using the standard array architecture, we constructed gRNA arrays of increasing lengths (3–16 gRNAs), targeting varying numbers of genes (Fig. 2E). While all gRNAs were able to mediate editing of ≥10% in at least one array, expressing gRNAs from an array generally lowered their editing frequencies. This effect was particularly pronounced for arrays composed of nine or more gRNAs, and specifically affected gRNAs located towards the 3'-end of the array (Fig. 2E and Supplementary Fig. 6). We

hypothesized that this effect was caused by a weak RNA Polymerase III (Pol-III) terminator sequence present in the DR of LbCas12a[20,21] that, if repeated many times, leads to premature termination of array transcription. We attempted to overcome this issue by expressing the gRNA array under the control of an RNA Polymerase II (Pol-II) promoter (Fig. 2A), a strategy utilized in other CRISPR-based assays[15,19]. Interestingly, we observed that the first seven gRNAs in the array achieved higher or equivalent editing frequencies when expressed from a Pol-III promoter (hU6) than with either Pol-II promoter (CMV or EFs1a). However, we made the reverse observation for gRNAs in position 8-16, for which the Pol-II-driven gRNA arrays mediated higher editing frequencies (Fig. 2F and Supplementary Fig. 6). In summary, we observed editing frequencies of up to 71.3 ± 2.4% for the Pol-III-driven array, but only eight gRNAs mediated efficiencies of >5%. In contrast, the CMV-promoter driven gRNA array reached a maximum editing

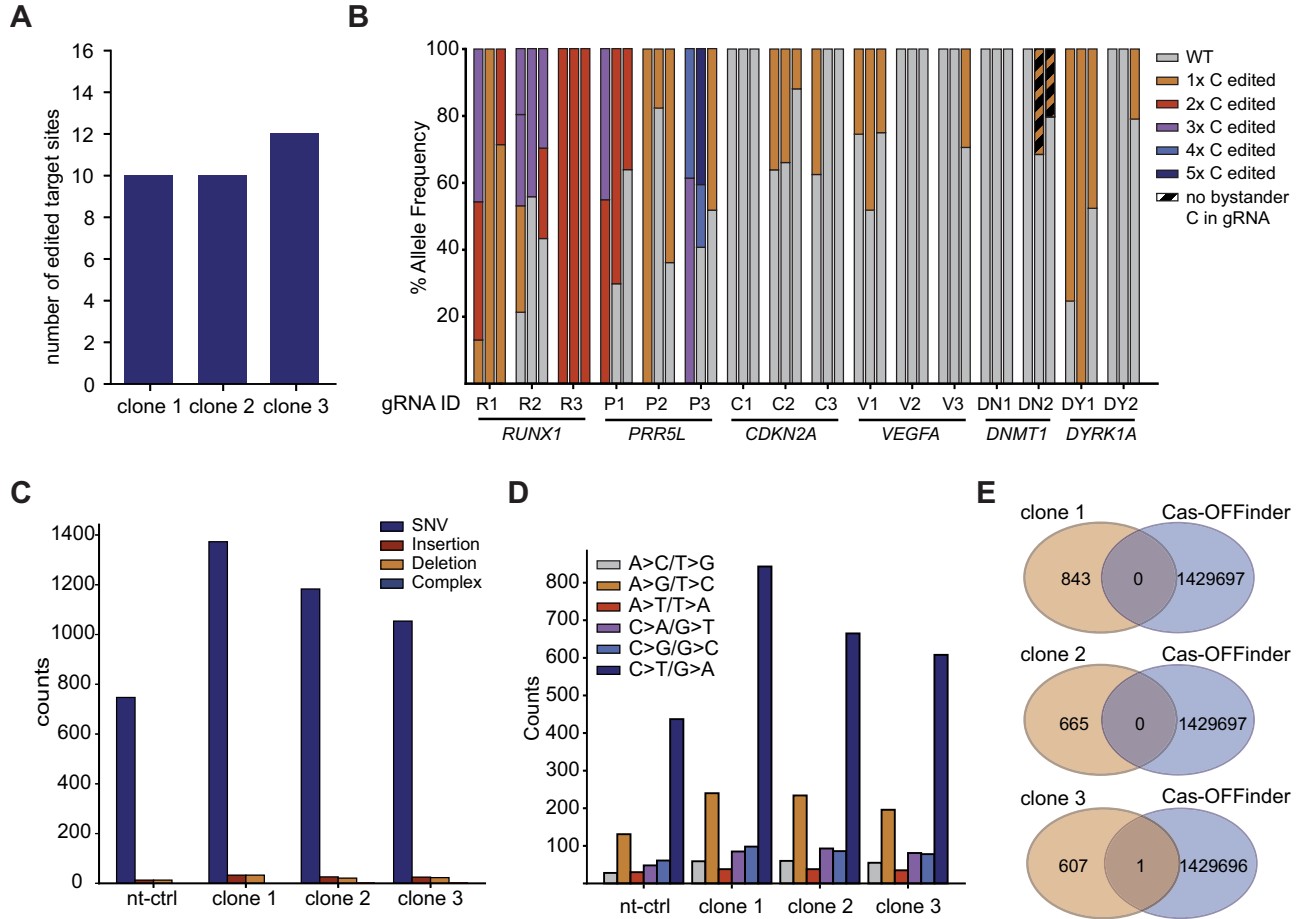

**Fig. 3 | Genome wide off-targets and bystander edits generated in BEACON2 treated cells. A** Total number of mutations found in three edited clones and one clone transfected with a non-targeting (nt-ctrl) gRNA. **B** Total number of detected single-nucleotide variants (SNVs) by base-change for three edited clones and one clone transfected with an nt-ctrl gRNA. **C** Summary of allele frequencies observed at the 16 targeted sites after editing with the 16x gRNA array (Fig. 2E), showing combined outcomes based on the number of edited Cs. Data represents three cell clones isolated from the edited population. **D** Total number of edited target sites per clone. **E** Overlap of C > T/G > A SNVs detected in our analysis and predicted off-target sites by Cas-OFFinder.

frequency of 62 ± 1.8%, but 15/16 gRNAs mediated an efficiency of above 5%, and 12/16 gRNAs reached efficiencies of over 10%. These data demonstrate the feasibility of MBE across multiple genes at high efficiency and highlight the importance of choosing an appropriate expression system for gRNA arrays depending on their length and composition.

## BEACON2-mediated multiplex base editing co-occurs with off-targets

Recent studies have investigated the mutagenic potential of Cas9-derived CBEs, identifying genome-wide SNV off-targets and BMs as the two major classes of unintended editing outcomes[6,22]. Because equivalent studies have not been performed for Cas12a-derived BE systems, we isolated clones from the cell population transfected with the hU6 promoter-driven 16x gRNA array (Fig. 2E) and from the population transfected with a non-targeting control gRNA (nt-ctrl) to perform whole genome sequencing (WGS). EditR[23] results revealed that four out of seven clones isolated from the 16x gRNA array population were edited at levels of 47-96% at the *RUNX1* (gRNA R1, R2, and R14) and *DYRK1A* (gRNA DY1) loci. Based on this data, we selected three clones that were submitted to WGS. We first measured the editing outcomes at the 16 gRNA target sites. Using our custom variant analysis workflow (see "Methods"), we found that clones 1 and 2 were edited at 10, and clone 3 at 12 of the targeted loci (Fig. 3A).

Notably, different clones displayed different editing outcomes when taking into consideration the number and position of converted bases. As such, gRNAs R1, R2, P1 and P3 mediated editing at different numbers of Cs in different clones with R2, R3 and P1 introducing BMs in all three clones (Fig. 3B). Interestingly, we observed only isolated edits without BMs with gRNA V1, however, the position of the edited C within the gRNA window was different for clone 2 (C9) than for clones 1 and 3 (C7) (Supplementary Fig. 7). Together, these data demonstrate MBE at a clonal level, supporting MBE observed in population-level assays.

Next, we sought to identify potential genome-wide off-target edits introduced by BEACON2. In all cell clones, we found that most detected mutations were SNVs (Fig. 3C) with an enrichment in C > T/ G > A mutations (Fig. 3D). This mutation signature has previously been identified in multiple cancer types as the mutational hallmark of APOBEC family cytidine deaminases, including APOBEC3A, a core component of BEACON2[24]. Only one SNV overlapped with predicted off-target sites (Fig. 3E), suggesting that the detected mutations are likely gRNA-independent but accumulate due to sustained expression of the CBE system, as other studies have reported for nCas9-derived BE systems[22,25]. Together, our WGS analysis confirms our ability to edit many sites across a single genome, while highlighting the need to develop strategies for overcoming inherent limitations such as DNA off-target deamination and BMs.

## Truncated gRNAs mitigate BEACON2-mediated bystander mutations

The use of BE systems for the study of trait-associated SNVs is limited by their inability to discriminate between multiple identical bases located within the activity window of the deaminase moiety. Consequently, not only the target base but also the surrounding bases may be modified, leading to BMs. A recent study investigating the dynamics of base editing suggests that BMs do not arise simultaneously with the edit at the main target base[26]. We hypothesized that this would allow us to exploit the mismatch sensitivity of Cas12a in two distinct ways. First, the introduction of intentional mismatches into the gRNA sequence has been shown to reduce bystander mutations in the context of Cas9-derived systems, motivating testing a similar approach for Cas12a-derived systems[27,28]. Second, Cas12a exhibits a uniquely high mismatch sensitivity when used with truncated gRNAs while retaining the ability to bind to the DNA target site[29,30]. We reasoned that the first base edit mediated by a truncated gRNA would create a mismatch between gRNA and target site, leading to Cas12a dissociating from the DNA and preventing rebinding. We first tested if introducing intentional mismatches in a full-length gRNA would be sufficient to mitigate BMs installed by BEACON2. Replacing the main target C of the previously validated gRNAs R1 and R7 with either a T or a G, lead to an overall reduction in editing frequency, but importantly mitigated BMs at surrounding nucleotide positions for at least one of the two mismatched gRNAs (Fig. 4A, B and Supplementary Fig. 8A, B).

Next, we tested the editing capability of 15 gRNAs targeting *RUNX1* on chromosome 21 after truncating them to a length of 15 nt (Fig. 4E and Supplementary Fig. 8B). While we observed no editing for six of the truncated gRNAs (e.g., R2, R3, R5, R9, R10, R16), the remaining truncated gRNAs mediated editing at varying frequencies at the target base with reduced BMs in the 20 nt window of the respective full-length gRNAs. We isolated three clones from the population transfected with the wildtype 15 nt gRNA R1 and R7 (Fig. 4A, B), to determine if the reduction in BMs on the population level would be reflected on a clonal level. Importantly, HEK293 cells, the parental cell line of HEK293-B2, have been shown to harbor four copies of chromosome 21[31]. Therefore, clonal editing outcomes are expected to range from 25–100% C-to-T conversion. Sanger sequencing revealed a gRNA R1 clone with an isolated 93% edit at C10, and two clones with isolated 61% and 70% edits at C10. For gRNA R7, we observed two clones with a 97% edit at C9 and 22–27% edit at C7 and C12 (Fig. 4C, D and Supplementary Fig. 8C, D). Lastly, we combined both approaches to mitigate BMs and introduced mismatches into the 15 nt version of R1 and R7, which led to a further decrease in BMs for R7 while maintaining editing at the target C but led to reduced or abolished editing at the target C for R1 (Fig. 4A, B and Supplementary Fig. 8A, B).

Knowing that truncated gRNAs can direct editing outcomes towards single-base-pair conversion, we sought to determine if we could use this strategy in conjunction with array-based gRNA expression to achieve editing of multiple target sites. We first combined the 15 nt wildtype gRNA R1 and R7 sequences into a hU6-driven double gRNA array and observed editing with low levels of BMs at both target sites (Supplementary Fig. 8E, F), confirming that BEACON2 can process gRNA arrays containing truncated gRNAs. Next, we combined six truncated gRNA wildtype sequences into two gRNA arrays, expressed under the control of the CMV promoter. The gRNA array v1 contained the six truncated gRNAs in ascending order of editing frequency at the main target C when expressed as a single gRNA, while the order was reversed for the gRNA array v2. In addition, we constructed the same arrays with the respective full-length versions of the gRNAs. For both array configurations, we observed editing at five out of the six targeted sites, with reduced BMs observed in all cases (Fig. 4F, G). However, editing frequencies were low for gRNA R15 and R11, independent of their position in the array, and gRNA R4 did not mediate editing when expressed from either array. Together, these data demonstrate that truncated gRNAs can efficiently mitigate BMs and can be expressed from gRNA arrays to facilitate precise MBE in mammalian cells.

## Multiplex base editing across human cell lines

The broad utility of MBE strategies for the generation of disease models or synthetic genomes depends on their ability to function across different cell lines that represent the phenotype or disease context of interest. Therefore, we tested BEACON1 and BEACON2 in combination with a triple gRNA array targeting *RUNX1* (gRNA R1, R2 and R14) across a panel of cell lines. Using protocol 2 (Supplementary Fig. 2A), we observed editing for both BE systems in all cell lines tested, though with varying editing frequencies (Fig. 5A and Supplementary Fig. 10). BEACON1 and BEACON2 mediated the highest editing frequencies of 60.1 ± 2.7% and 59.4 ± 2.9% in HeLa cells, respectively. In contrast, A375, HT1080, and U2OS cells were edited at frequencies of up to 14.8 ± 2% (HT1080, gRNA R1). In all cell lines, gRNA R2 performed worse than gRNA R1 and R14, and BEACON1 mediated higher editing frequencies than BEACON2. Encouraged by these results, we chose to test BEACON1 and BEACON2 with the 16x gRNA array expressed under the control of the CMV promoter in HeLa cells (Fig. 5B and Supplementary Fig. 11). Despite the need to co-transfect the gRNA expression plasmid and the BE system expression plasmid, we observed comparable editing frequencies in HeLa cells transfected with BEACON2 as in HEK293-B2 cells. Notably, some target loci were edited at higher frequencies in HeLa cells transfected with BEACON1 (e.g., gRNA R2, V1, and V2), while others were edited at lower frequencies (e.g., gRNA P2, V3, and DN2). Taken together, these results demonstrate the applicability of our MBE strategy to multiple cell lines and motivate the optimization of cell line-specific protocols in future studies to further enhance editing frequencies. In addition, these results open up the possibility to generate complex synthetic genomes or reconstruct disease genotypes in disease model cell lines.

## Discussion

Creating many precise edits across a human genome is non-trivial and remains a defining challenge in the field of genome engineering, hindering comprehensive studies of complex genotype-phenotype relationships. Motivated by this challenge, this study screened six Cas12a-derived BE systems for their capability to mediate MBE and evaluated multiple gRNA array architectures for the efficient expression and processing of long gRNA arrays. Ultimately, we achieved editing at up to 15 target sites across 6 genes using BEACON2 and a single gRNA array expression cassette in HEK293-B2 and HeLa cells. In addition, we demonstrate that using BEACON2 with gRNA arrays composed of truncated gRNAs reduces BM frequencies at some target sites, permitting MBE with greater precision than can be achieved with full-length gRNAs.

We evaluated multiple gRNA array architectures and demonstrated that flanking each gRNA within an array with a SynSep can significantly impact gRNA performance. In contrast to previous reports, we found that this effect is independent of the standard SynSep sequence AAAT, suggesting a more complex relationship between gRNA array composition, SynSeps, and array processing[19]. These results motivate future studies of SynSeps and their impact on the secondary and tertiary structure of gRNA array transcripts. We envision that the use of computational models that can accurately predict the structure of gRNA arrays may allow for the design of optimal gRNA array sequences, taking into consideration the order of gRNAs within the array, and predicting variable SynSeps that aid the optimal folding and processing of gRNA arrays. We further demonstrated that the expression of short gRNA arrays from a Pol-III promoter is effective, but long arrays benefit from Pol-II-driven expression for robust editing of all target sites. Consequently, future expression systems may be designed to express both the BE system and the gRNA array from the same Pol-II-driven promoter, thereby reducing plasmid size and eliminating the

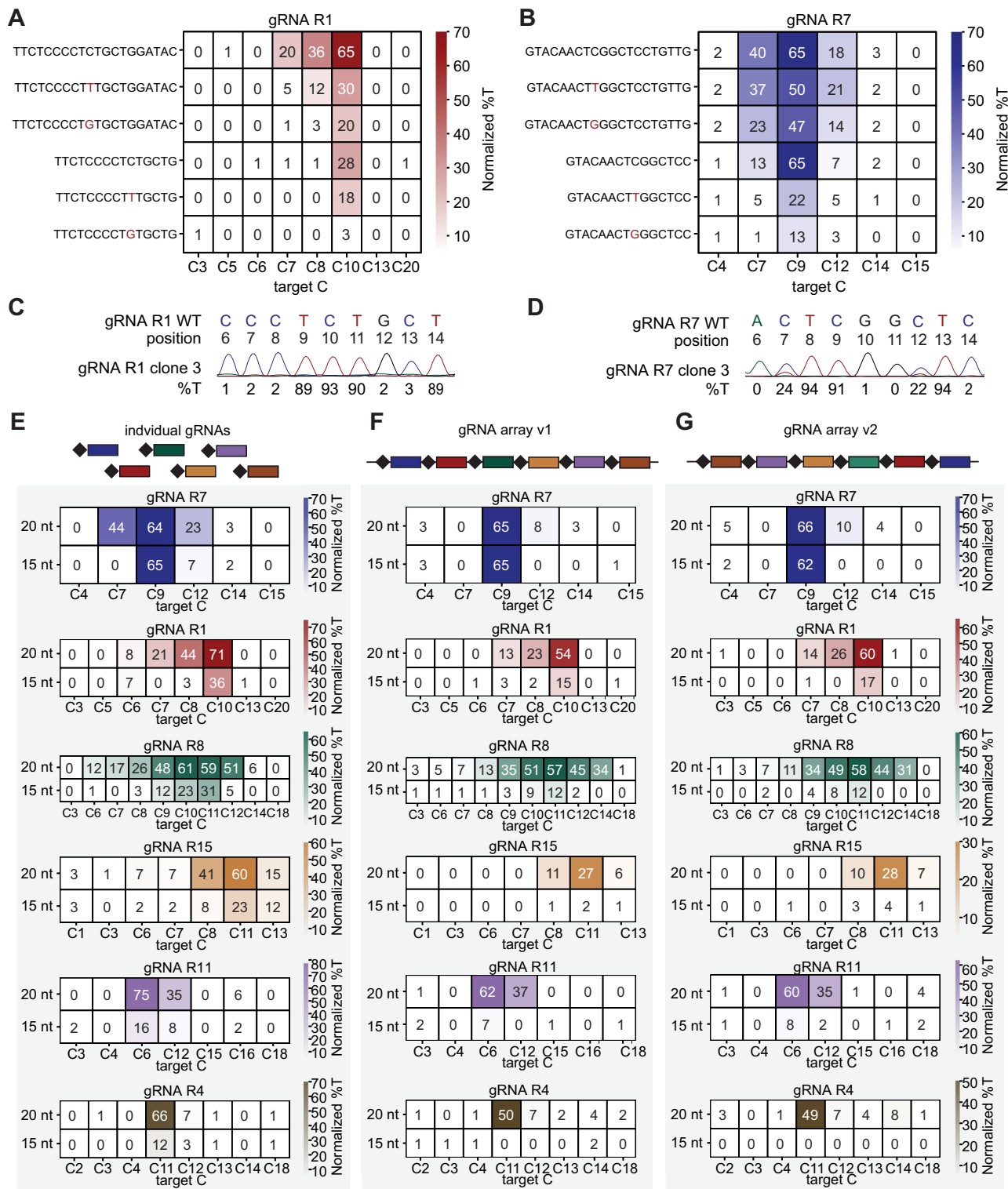

**Fig. 4 | Engineered gRNAs reduce frequencies of bystander mutations mediated by BEACON2. A, B** Editing outcomes of different gRNA designs for gRNA R1 and R7. **C, D** Sanger sequencing traces and %T values of a cell clone picked from the 15 nt wildtype sequence condition shown in (**A** and **B**). **E** Editing outcomes of six *RUNX1* targeting gRNAs when expressed as either individual 20 nt or individual 15 nt gRNAs under the control of the CMV promoter. **F, G** Editing outcomes of the same six gRNAs when expressed as an array of 20 nt or 15 nt gRNAs. The order of gRNAs in array v2 in (**G**) is reversed from gRNA array v1 in (**F**). **A, B, E–G** show normalized mean %T values. Normalization was performed by subtracting the mean %T of the non-targeting (nt-ctrl) condition from the mean %T of the experimental condition. Source data are provided as a Source Data file.

need for co-transfections. We expect that these findings are applicable to all CRISPR genome engineering approaches that utilize gRNA arrays, emphasizing the need to consider promoter choices carefully when designing expression systems.

Potential off-target edits, including genome-wide mutations as well as BMs within the gRNA window, represent two critical challenges when developing genome engineering strategies. While previous studies have demonstrated considerable off-target DNA edits for Cas9-

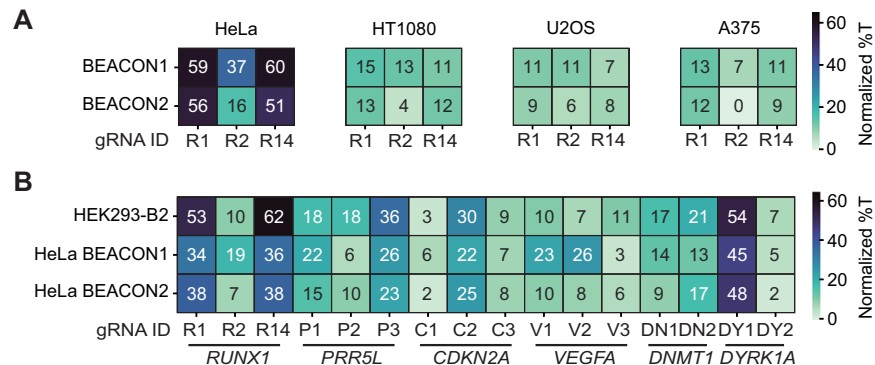

**Fig. 5 | Multiplexed base editing in diverse cell lines. A** Editing outcomes mediated by BEACON1 and BEACON2 when used with a triple gRNA array targeting *RUNX1*. **B** Editing outcomes mediated by BEACON1 and BEACON2 at 16 target sites in HeLa cells transfected with a CMV promoter-driven 16x gRNA array and the respective BE system. Editing outcomes in HEK293-B2 cells mediated by BEACON2 transfected with the same 16 × gRNA array as presented in Fig. 2F. **A, B** show normalized mean %T values at the highest edited C for each gRNA as determined based on the single gRNA condition. Normalization was performed by subtracting the mean %T of the non-targeting (nt-ctrl) condition from the mean %T of the experimental condition. Source data are provided as a Source Data file.

derived BE systems[6,22], similar studies have not yet been conducted on dCas12a-derived systems. While BEACON2 has previously been shown to induce only basal levels of RNA off-target mutations[12], our data demonstrate that BEACON2 mediates high DNA off-target editing frequencies that are gRNA-independent and thus do not overlap with predicted off-target sites. In this regard, next-generation MBE strategies will likely need to employ more transient or conditional expression schemes than the ones utilized in this study to reduce the number of off-target edits observed in a single cell. In addition, we demonstrated that the use of truncated gRNAs with a dCas12a-derived BE system like BEACON2 can reduce BMs that occur in the gRNA window of the full-length gRNA. Although editing outcomes are variable across the gRNAs tested, we found examples of gRNAs that mediate precise base-pair changes when truncated.

This study motivates screens of large, truncated gRNA libraries to deduce generalizable gRNA design rules and the development of predictive tools that inform the optimal gRNA design when attempting to install multiple, precise nucleotide changes into a genome. Further, the use of truncated gRNAs with genome engineering systems derived from other Cas variants should be explored to determine changes in editing outcomes and the generalizability of this strategy. Our findings pave the way for comprehensive studies of complex phenotypes and the rapid generation of disease models or synthetic genomes[7].

## Methods

### gRNA expression vector construction

The gRNA expression vector pRC167 containing the AsCas12a direct repeat sequence was gifted to us by the Chen laboratory and used to construct phU6 (Supplementary Fig. 4A) containing the LbCas12a direct repeat sequence, using Gibson Assembly with NEBuilder HiFi DNA Assembly (New England Biolabs #E2621). The gRNA expression plasmids pCMV and pEFs1a were designed according to the design in Magnusson et al.[19] and constructed as a Golden Gate assembly with NEB BbsI-HF (New England Biolabs #R3539S) from PCR fragments and gene fragments (Twist Bioscience) with the appropriate overhangs.

Single, double, and triple guide arrays were ordered as two overlapping DNA oligonucleotides or Ultramers (IDT) with appropriate four-base pair overhangs, annealed and cloned into pRC167 or pAYS009 using the BsmBI-v2 NEBridge Golden Gate Assembly Kit (New England Biolabs #E1602L) following the manufacturer's instructions. Arrays containing more than three gRNAs were either ordered as gene fragments or clonal genes (Twist Bioscience) flanked by BsmBI

cut sites or split up into multiple (< 200 bp) fragments and ordered as single-stranded DNA Ultramers (IDT). If ordered as Ultramers, the top strand of the array was assembled using short bottom oligonucleotides, as described in Cooper & Hasty[32]. The resulting ssDNA fragment was utilized as a PCR template to amplify arrays of different lengths and fill in the second strand of the array. To allow subsequent cloning into pAYS009, pAYS205, and pAYS206, all forward and reverse primers encoded a BsmBI cut site. In addition, the reverse primer contained the sequence of the last spacer within each array.

### gRNA and gRNA array design

Cas12a guides were designed using ChopChop[33] with a TTTV PAM sequence and a length of 20 nt, unless otherwise noted. For Fig. 2C, a total of 26 gRNAs targeting *RUNX1* were tested, and 14 out of 16 gRNAs that mediated efficient base editing were selected to construct the shown arrays. For Fig. 2E, 3 gRNAs per target gene were designed and tested, and all gRNAs that mediated efficient base editing were combined into the shown array. The non-targeting gRNAs for Supplementary Fig. 3C were taken from Magnusson et al.[19]. The gRNA array expression cassettes were designed by alternating DR sequences and validated gRNAs, or DR sequences, synthetic separator sequences, and validated gRNAs. Synthetic separator sequences were either fixed (AAAT), as published in Magnusson et al.[19], or variable. To design variable synthetic separator sequences, the dot bracket notation of the desired gRNA array structure and the gRNA array sequence containing four degenerate nucleotides as synthetic separator sequences was used as input for the NUPACK design function[34].

### Mammalian cell culture

HEK293 cells were a gift from the Rinehart laboratory and used as the parental cell line to generate HEK293-B2 cells as described below. HeLa cells were a gift from the MacMicking laboratory. U2OS (ATCC #HTB-96) and HT1080 (ATCC #CCL-121) cells were purchased from ATCC. A375 cells were a gift from the Chen laboratory. All cell lines were grown at 37 °C in 5% CO$_2$ in Dulbecco's modified Eagle's medium (DMEM) supplemented with 10% (v/v) fetal bovine serum (VWR) and 1% (v/v) penicillin and streptomycin (Gibco #1965092). Cells were passaged upon reaching 80–90% confluency and regularly tested for mycoplasma contamination using a PCR Mycoplasma detection kit (Thermo Fisher Scientific # J66117).

### Generation of stable cell line HEK293-B2

The BEACON2 expression cassette from pCMV-BEACON2[12] (Addgene ID171698) was cloned into pSBbi-Hyg[35] (Addgene ID 60524) using

Gibson assembly. Both pSBbi-Hyg and pCMV(CAT)T7-SB100[35] (Addgene ID 34879) were a gift from the Rinehart lab. HEK293 cells were seeded at $1 \times 10^6$ cells per well in 6-well plates. After 24 h, cells were transfected with 500 ng of cargo plasmid, 50 ng of SB100X, and 450 ng of salmon sperm DNA (Thermo Fisher Scientific #15632011). Transfections were done using Lipofectamine 3000 (Thermo Fisher Scientific #L3000015), following the manufacturer's instructions. The following day, the media was replaced with DMEM containing Hygromycin (Mirus #MIR5930S) at a concentration of 200 µg/mL, and cells were cultured and selected for 15 days, until all cells in an untransfected control well had died. Subsequently, cells were expanded and stocked for subsequent experiments.

### Isolating clonal cell lines
To generate clonal cell populations, polyclonal cell pools were seeded at a very low density (approximately 60 cells per 10 cm culture dish) to obtain well separated colonies formed from single founder cells. Once colonies had formed, they were manually picked of the plate expanded, and used for subsequent experiments.

### Transient transfection of base editing plasmids
On day 0, HEK293 and HEK293-BE2 cells were seeded in 24-well plates at a density of $1.4 \times 10^5$ cells per well. HeLa cells were seeded at a density of $3 \times 10^4$, HT1080 cells were seeded at a density of $6 \times 10^4$, A375 cells were seeded at a density of $4 \times 10^4$, and U2OS cells were seeded at a density of $0.4 \times 10^5$ cells. After 24 h, cells were transfected with 50 µl Opti-MEM (Gibco #31985070) containing 2 µl of Lipofectamine LTX and 0.5 µl Plus reagent (Thermo Fisher Scientific #15338030) and the respective gRNA and base editor plasmids. For co-transfection experiments, HEK293 cells were transfected with 340 ng gRNA expression plasmid and 500 ng base editor plasmid (Addgene IDs 171697, 171698, 138504, 138506, 114081, and 114082, Supplementary Data 1). HEK293-B2 cells were transfected with 800 ng gRNA expression plasmid only. The following day, cells were transferred into DMEM containing puromycin (Gibco #A1113803) at a concentration of 2 µg/mL. Cells were transferred again 72 h post-transfection and cultured without puromycin for an additional 72 h in 12-well plates. Cells were harvested on day 7 by adding 100 µl of 0.25% Trypsin-EDTA (Gibco #25200056) to each well, transferring the cell suspension to PCR tubes, and boiling the cells at 95 °C for 20 min. Alternatively, QuickExtract (Lucigen #QE0905T) solution was used according to the manufacturers protocol. Cell lysates were subsequently used as PCR templates for the amplification of targeted genomic loci or stored at −20 °C until further use.

### Targeted PCR amplification, Sanger sequencing and data analysis
The genomic region surrounding each target locus was amplified using primers listed in Supplementary Data 1 and KAPPA HiFi DNA polymerase (Roche #KK2602) following the manufacturers protocol. Specific amplification was verified by DNA electrophoresis prior to purification of PCR products using SpinSmart PCR Purification and Gel Extraction Columns (Denville #CM500250). Purified PCR products were sent for Sanger sequencing (Quintara Biosciences), and results were analyzed using either the EditR web application or an automated version of EditR[23]. Since the web application associated with EditR can only analyze one sample at a time, we wrote a custom R script to automate and increase the consistency of the EditR analysis across samples. This script introduces a method to automatically trim the noisy ends of the Sanger traces, a step that is a manual process in the EditR web application. Specifically, it trims each end up until the noise (the sum of the values corresponding to non-basecall nucleotides) at each end is below a set threshold.

### Whole genome sequencing
For whole genome sequencing, genomic DNA (gDNA) was isolated from seven cell clones using the Dneasy Blood and Tissue (Qiagen #69504) following the manufacturer's instructions for RNA-free gDNA extractions. PCR amplification of the *RUNX1* and *DYRK1A* target loci was performed as described, and EditR was used to determine editing levels for each clone. Out of the seven clones, four were edited at the analyzed sites. Based on these results, we picked three clones to submit for WGS. Library generation and sequencing using Illumina Novaseq X were carried out at the Yale Center for Genome Analysis (YCGA).

### Variant calling
Variant calling was performed by the YCGA. The variant calling pipeline uses the best practice GATK 4[36] method to call variants, first aligning the sequencing using BWA MEM[37] to the hg38 human reference plus decoy sequences. It then marks PCR duplicates using Picard's MarkDuplicates command. The GATK 4.1.2.0 software[38] is used to recalibrate base quality scores and to generate CRAM and GVCF files for each sample. Once GVCF files have been generated, joint variant calling is performed, and variants are filtered using GATK's variant quality score recalibration.

### Analysis of on-target edits
To determine the on-target efficiency at an individual chromosomal position located within one of the 16 gRNA sequences, we extracted the relevant variant calls with a mapping quality score ≥ 35 and a read depth of ≥ 10 from the VCF file and plotted the allele frequency as determined by the variant caller. To determine the frequency of BMs and the editing outcomes at each target site with higher sensitivity, we used the alignment files directly and, for each on-target site, extracted all reads covering the entirety of the 20 nt gRNA sequence with a mapping quality score ≥ 35. Next, we removed any alleles that were supported by < 4 total reads. The frequency of each allele was then calculated based on the total number of reads covering the gRNA window and the number of reads supporting the occurrence of each individual allele.

### Analysis of off-target edits
To identify de novo off-target mutations, any variant called across all four samples was considered a variant of the parental cell line and removed from the subsequent analysis. Variants called in the edited clones for which there was no information on the status of the nt-ctrl sample were also removed from the analysis. We further only considered variants with an MQ ≥ 40 and read depth > 10 to be reliable in the analysis. In addition, we removed any variants that fell into low complexity regions, as variants cannot be called reliably. This was done using low complexity regions defined by the Genome-in-a-bottle (GIAB) consortium[39]. Any variant that passed the described analysis pipeline was considered in the subsequent analysis. Potential off-targets of the used gRNAs were predicted using Cas-OFFinder with a TVVV PAM sequence, five possible mismatches, and a DNA and RNA bulge size of two or less[40]. To identify the overlap between de novo variants and predicted off-targets of the utilized Cas12a gRNAs, the 20 nt located 3' of the PAM sequence were considered.

### Statistics and reproducibility
All data sets were generated with at least three replicates unless specified otherwise, and error bars are reported as mean ± SD. No statistical method was used to predetermine sample size, and no data was excluded from analysis. The authors were not blinded to allocation during experiments and outcome assessment. The p-values presented were calculated using Turkey's Honest Significant Difference test for multiple comparisons, with a confidence level cutoff of ns, not

significant ≥ 0.05, *$p < 0.05$ **$p < 0.01$, ***$p < 0.001$, ****$p < 0.0001$. Statistical analyses were performed on raw data values that can be found in the Source Data File and Supplementary Figs. 5 and 8. Only the preferred target Cs as defined in Supplementary Data 1 were considered, unless otherwise specified.

## Reporting summary

Further information on research design is available in the Nature Portfolio Reporting Summary linked to this article.

## Data availability

All data supporting the findings reported in this study and all DNA sequences used in this study are available within the manuscript and its Supplementary Information. Raw fastq files of the WGS experiment performed in this study have been deposited in the NCBI Sequence Read Archive (SRA) under project accession code PRJNA1120172. The source data of all figures is provided in a Source Data File. All DNA sequences are provided in Supplementary Data 1. Source data are provided in this paper.

## Code availability

The code used to automate the EditR analysis of Sanger Sequencing files in batch mode is available at https://github.com/etowahadams/EditRBatch and archived on Zenodo (https://doi.org/10.5281/zenodo.15060672). The code is licensed under the GNU General Public Licence (GPL).

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

## Acknowledgements

We are grateful to members of the Isaacs and Rinehart laboratories for their critical discussions, feedback, and for reviewing the manuscript. We thank the Sidi Chen laboratory for gifting us plasmid pRC167 and the A375 cell line, the Jesse Rinehart laboratory for gifting us the HEK293 cell line, and the John MacMicking laboratory for gifting us the HeLa cell line. We thank Vladimir Chituc for his help with the statistical analysis of our data. We thank Ben Rosenbluth for his help cloning plasmids used in this study. Research reported in this publication was supported by Yale University, the National Institute of General Medical Sciences of the National Institutes of Health (GM117230 to F.J.I.) and the Carlsberg Foundation [CF22-1046 to M.T.N.].

## Author contributions

A.Y.S. and F.J.I. conceived of and led the design of the study. A.Y.S. performed experiments and wrote the paper, incorporating input from all authors. E.W.A. conceived of and wrote the script used to analyze Sanger sequencing files in batch mode. M.L. designed the WGS analysis workflow and contributed to the interpretation of off-target results. M.T.N. proposed the idea and designed variable synthetic separators based on predicted secondary structures of arrays.

## Competing interests

The authors declare no competing interests.
