## [Transparent Peer Review file · Nature Communications]

Precision multiplexed base editing in human cells using Cas12a-derived base editors.

Corresponding Author: Professor Farren Isaacs

This manuscript has been previously reviewed at another journal. This document only contains information relating to versions considered at Nature Communications. Mentions of the other journal have been redacted.

Version 0:

Reviewer comments:

Reviewer #1

(Remarks to the Author)

The authors have addressed my concerns in this revised submission. The added experiments examining truncated guides in the context of a larger array and the investigation of multiple cell lines add to the rigor of the work. Although the resultant editing efficiencies from these experiments appears to be reduced, this can be a point for future optimization.

(Remarks on code availability)

I am not versed with R and therefore cannot review the code's validity. However, there is a README file with instructions and example data for analysis.

Reviewer #2

(Remarks to the Author)

* General comments:

Base editors, such as cytosine base editors (CBEs) and adenine base editors (ABEs), are powerful tools capable of creating precise base conversion with generating little DNA double strand breaks. In this study, the authors optimized protocols and demonstrated multiplexed base editing in human cells using various Cas12a-derived base editors. The authors conducted multiple targeting at 14 different sites in the genome thanks to the ability of Cas12a to process crRNA arrays itself. The authors also used pol II promoter for stable expression of the long sgRNA array and used truncated crRNAs for reduced bystander edits and higher specificity. As one of the referees of this study in another journal, this reviewer still believes that the main idea of this study is a combination of previous results, but it can provide practical aspects of the dCas12a-CBE or ABE platform to a wide audience. I would like to raise some issues to strengthen this study.

* Specific comments:

1. To my knowledge, Xiao Wang et al. [PMID: 32492431] previously developed dCas12a-based CBE variants and firstly named them as BEACON (base editing induced by human APOBEC3A and Cas12a without DNA break) or BEACON2. However, I think many readers are not familiar with these tools or names, so it would be better to explain them in detail in the main text.
2. Previously, David Liu group first developed dCas9-based CBE (named ver1) which displayed very low base editing efficacy. Thus, they further improved the base editing efficiency by using Cas9 nickase (D10A) instead of dCas9, which made a nick at the gRNA binding strand opposite to the deamination strand increasing the editing activity. However, for Cas12a effectors, it is not easy to make a nickase format because the Cas12a contains only one RuvC-like nuclease domain. I am curious how dCas12a-BE showed such higher editing rates, compared to dCas12-based BE platforms. It should be discussed.
3. According to the protocols in Supplementary Fig. 2, it seems that the authors basically used puromycin-selection system. But, in some previous results, they did not use any selection system. Therefore, it is unfair to claim that the optimized version of dCas12a-BE demonstrated in this study exhibited superior base editing activity than other previously developed

platforms.

In addition, considering the further in vivo applications in animals where selection systems would not be working, it is necessary to provide dataset without any selection systems.

4. In Figure 3, it is very nice to analyze the genome-wide off-target effects in BEACON2 treated cells. In addition to this, it would be better to discuss any off-target effects in RNA transcripts, too. It can be addressed by RNA-seq or NGS with RNA cDNAs.

(Remarks on code availability)

Reviewer #3

(Remarks to the Author)

The authors comprehensively addressed to my comments that I provided when the manuscript was considered for [Redacted]. I appreciate their hard work in revising the paper.

(Remarks on code availability)

Version 1:

Reviewer comments:

Reviewer #2

(Remarks to the Author)

The authors have mostly answered the issues I raised in the earlier review. However, I was disappointed that the authors did not highlight any of the changes in the main text, which was so uncomfortable.

(Remarks on code availability)

REVIEWER COMMENTS

Reviewer #1 (Remarks to the Author):

The authors have addressed my concerns in this revised submission. The added experiments examining truncated guides in the context of a larger array and the investigation of multiple cell lines add to the rigor of the work. Although the resultant editing efficiencies from these experiments appears to be reduced, this can be a point for future optimization.

- We thank the reviewer for the positive assessment of our revised manuscript.

Reviewer #1 (Remarks on code availability):

I am not versed with R and therefore cannot review the code's validity. However, there is a README file with instructions and example data for analysis.

Reviewer #2 (Remarks to the Author):

* General comments:

Base editors, such as cytosine base editors (CBEs) and adenine base editors (ABEs), are powerful tools capable of creating precise base conversion with generating little DNA double strand breaks. In this study, the authors optimized protocols and demonstrated multiplexed base editing in human cells using various Cas12a-derived base editors. The authors conducted multiple targeting at 14 different sites in the genome thanks to the ability of Cas12a to process crRNA arrays itself. The authors also used pol II promoter for stable expression of the long sgRNA array and used truncated crRNAs for reduced bystander edits and higher specificity. As one of the referees of this study in another journal, this reviewer still believes that the main idea of this study is a combination of previous results, but it can provide practical aspects of the dCas12a-CBE or ABE platform to a wide audience. I would like to raise some issues to strengthen this study.

* Specific comments:

1. To my knowledge, Xiao Wang et al. [PMID: 32492431] previously developed dCas12a-based CBE variants and firstly named them as BEACON (base editing induced by human APOBEC3A and Cas12a without DNA break) or BEACON2.

However, I think many readers are not familiar with these tools or names, so it would be better to explain them in detail in the main text.

- We thank the reviewer for raising this point. The nomenclature used to describe the different base editing systems throughout the manuscript is indeed derived from the respective publications that first developed those systems. To improve the clarity of the manuscript, we have introduced the full-length names of the base editing systems BEACON1 and BEACON2 before switching to the abbreviated names (lines 100-102).

2. Previously, David Liu group first developed dCas9-based CBE (named ver1) which displayed very low base editing efficacy. Thus, they further improved the base editing efficiency by using Cas9 nickase (D10A) instead of dCas9, which made a nick at the gRNA binding strand opposite to the deamination strand increasing the editing activity. However, for Cas12a effectors, it is not easy to make a nickase format because the Cas12a contains only one RuvC-like nuclease domain. I am curious how dCas12a-BE showed such higher editing rates, compared to dCas12-based BE platforms. It should be discussed.

- This is an interesting question and similar to one raised by Reviewer #1 during the first round of review. Based on the framing of the question, we infer that the reviewer is curious to see how dCas12a-BE systems show higher editing rates than dCas9-based BE platforms. Similar to our response to Reviewer #1 from the first round of review, we appreciate the insightful comments and agree with their assessment that no Cas12a nickase (nCas12a) that cleaves the target strand of the DNA has been reported in the literature. Although we agree that a base editing system derived from nCas12a could be more efficient than the systems tested, variants used in this manuscript have previously been shown to exceed editing efficiencies of 50% (the theoretical limit of a base editing system derived from a catalytically dead CRISPR nucleases (Komor et al., 2016 doi:10.1038/nature17946) and have been benchmarked against second and third generation Cas9-derived base editing systems (Wang et al., 2020) doi:10.1016/j.celrep.2020.107723). Considering that Cas9 and Cas12a utilize different PAM sequences, only few edits can be generated using either nuclease. Building off this recent literature, we have therefore focused our efforts on a comprehensive comparison of existing dCas12a-derived base editing systems, which is missing from the current base editing literature, rather than a comparison to existing Cas9-derived systems.

To improve the clarity of the manuscript, we are now explicitly mentioning the findings of the referenced study in the main text of the manuscript (line 75-77):

“Although most published BEs utilize a Cas9 nickase (nCas9), BEs utilizing a catalytically dead Cas12a (dCas12a) protein have been developed and shown to mediate similar or higher editing frequencies when compared to nCas9-derived BE systems^{12–15}.”

3. According to the protocols in Supplementary Fig. 2, it seems that the authors basically used puromycin-selection system. But, in some previous results, they did not use any selection system. Therefore, it is unfair to claim that the optimized version of dCas12a-BE demonstrated in this study exhibited superior base editing activity than other previously developed platforms.

- The reviewer raises an interesting point, similar to one raised by reviewer 1 during the first review. Like our response to Reviewer #1 from the first round of review, previous reports of dCas12a-derived base editing systems have employed different protocols to assess editing efficiencies, making comparisons across studies impossible. We therefore aimed to assess the editing efficiencies of six published systems using the same protocol, enabling direct comparisons between tested CBE, and tested ABE systems. To determine the editing efficiency of each system, rather than the combined effect of transfection efficiency and editing efficiency, we introduced the antibiotic selection described by the reviewer. Notably, we do not compare the editing efficiencies achieved in this manuscript to previous reports. As such, we believe that the introduction of the antibiotic selection in protocol 2 was a necessary addition to the protocol to accurately determine the editing efficiencies of the tested systems.

In addition, considering the further in vivo applications in animals where selection systems would not be working, it is necessary to provide dataset without any selection systems.

- To provide a comprehensive resource, the manuscript includes two datasets that were collected without selection (Supplementary Figure 1 and Supplementary Figure 2C-E), in addition to the datasets collected with protocol 2 which uses a puromycin selection. While we agree that the puromycin selection system presents a bottleneck for the use of the presented base editing strategy in an *in vivo* system, it significantly improved editing frequencies in our experiments. Importantly, we would like to point out that similar concerns were raised by this reviewer in the first round of revisions (comment 4,9 and 29) and would like to

clarify again, that the presented genome engineering method is not intended to be used in a therapeutic or *in vivo* context. Instead, our advances have utility in generating many precise edits in cell lines for functional genetic testing, the generation of synthetic genomes, or other applications requiring many edits.

4. In Figure 3, it is very nice to analyze the genome-wide off-target effects in BEACON2 treated cells. In addition to this, it would be better to discuss any off-target effects in RNA transcripts, too. It can be addressed by RNA-seq or NGS with RNA cDNAs.

- While the question of RNA off-target effects caused by BEACON2 is interesting, it has been addressed in a prior study by Wang et al. (2020). Out of four base editing systems tested in this study (BE3, AncBE4max, BEACON1 and BEACON2), BEACON2 was the only system for which the frequency of C to U mutations detected at the RNA level was similar to background levels. This finding was a factor that guided the design and execution of our study, which we now state in the main text of the manuscript (lines 154-159 and 360-263). In this manuscript, we therefore prioritized the investigation of off-target effects caused by BEACON2 on the DNA level, which have not yet been determined in the literature. Because the focus of the presented study was to establish a framework for the generation of complex genotypes in human cells using Cas12a-derived base editing systems, and because the question of RNA off-targets has been investigated in a prior study (Wang et al. (2020)), we believe it is beyond the scope of this manuscript and could be further investigated in future work.

Reviewer #3 (Remarks to the Author):

The authors comprehensively addressed to my comments that I provided when the manuscript was considered for **[Redacted]**. I appreciate their hard work in revising the paper.

- We thank the reviewer for the positive assessment of our revised manuscript.

REVIEWERS' COMMENTS

Reviewer #2 (Remarks to the Author):

The authors have mostly answered the issues I raised in the earlier review. However, I was disappointed that the authors did not highlight any of the changes in the main text, which was so uncomfortable.

- We thank the reviewer for their time and appreciate their recognition that we addressed issues in the prior versions of the manuscript.